# Moderate greenhouse climate and rapid carbonate formation after Marinoan snowball Earth

Lennart Ramme [1,2] ✉, Tatiana Ilyina[1,3,4] & Jochem Marotzke [1,3]

When the Marinoan snowball Earth deglaciated in response to high atmospheric carbon dioxide ($CO_2$) concentrations, the planet warmed rapidly. It is commonly hypothesized that the ensuing supergreenhouse climate then declined slowly over hundreds of thousands of years through continental weathering. However, how the ocean affected atmospheric $CO_2$ in the snowball Earth aftermath has never been quantified. Here we show that the ocean's carbon cycle drives the supergreenhouse climate evolution via a set of different mechanisms, triggering scenarios ranging from a rapid decline to an intensification of the supergreenhouse climate. We further identify the rapid formation of carbonate sediments from pre-existing ocean alkalinity as a possible explanation for the enigmatic origin of Marinoan cap dolostones. This work demonstrates that a moderate and relatively short-lived supergreenhouse climate following the Marinoan snowball Earth is a plausible scenario that is in accordance with geological data, challenging the previous hypothesis.

The Neoproterozoic (1000–541 million years ago, Ma) saw probably the most extreme climatic consequences of carbon cycle excursions in the Earth's history, including two long-lived periods of global glaciation that are commonly known as the Sturtian and the Marinoan snowball Earth[1–3]. When the Marinoan snowball Earth deglaciated in response to increased atmospheric $CO_2$ concentrations[4–7], the Earth must have subsequently transitioned into a much warmer supergreenhouse climate[8–10]. This rapid change posed a major burden for the early forms of life that developed prior to the Marinoan snowball Earth and must have survived also the deglaciation and the ensuing supergreenhouse climate[11–13]. The severity and duration of the supergreenhouse climate are therefore important factors for reconstructing the evolution of life during that time. However, many uncertainties come with the conditions at the start of the deglaciation, and some of the unknown quantities are important regulators of Earth's climate, such as the amount of carbon in the atmosphere and the ocean[2,14,15]. Based on the large uncertainties in those quantities, one could speculate that also the

evolution of the climate in the snowball Earth aftermath should be uncertain. By contrast, the existing literature seems in agreement that the atmospheric $CO_2$ concentration, and hence the supergreenhouse climate, declined over a long time scale of up to millions of years, driven by the strong chemical weathering of continental rocks drawing down atmospheric $CO_2$[2,3,8,9,16]. But in addition to this slow process of continental weathering, the large climatic shifts will also have induced changes in the physical, chemical and biological dynamics of the ocean, initiating faster carbon cycle processes that act on a time scale of less than ten thousand years. The apparent certainty about the slow decline of the supergreenhouse climate is simply a result of the fact that the faster carbon cycle processes have never been quantified. We fill this gap by simulating a period of five thousand years after the Marinoan snowball Earth with a comprehensive Earth system model ICON-ESM[17] that resolves the co-evolution of the climate, the ocean and the carbon cycle, while taking existing uncertainties into account.

[1]Max Planck Institute for Meteorology, Hamburg, Germany. [2]International Max Planck Research School on Earth System Modelling, Hamburg, Germany. [3]Center for Earth System Research and Sustainability (CEN), Universität Hamburg, Hamburg, Germany. [4]Helmholtz-Zentrum Hereon, Geesthacht, Germany. ✉e-mail: lennart.ramme@mpimet.mpg.de

## Results

### Uncertain post-snowball Earth conditions

The major uncertainties that influence the evolution of the air-sea $CO_2$ exchange in the snowball Earth aftermath are: (1) The amount of meltwater that enters the ocean during the deglaciation being between 500 and 1500 m of sea-level equivalent[18–21]. (2) The atmospheric $CO_2$ concentration that triggered the deglaciation is only loosely constrained to $10^4 – 10^5$ parts per million ppm,[4–7]. (3) The ocean carbon reservoir could have either been in equilibrium with atmospheric $CO_2$ or the ocean was isolated from the atmosphere by the sea-ice cover during the snowball Earth, offering the possibility of a strong disequilibrium[3,14]. This depends on the meridional extent of the ice cover, that is, on whether the Marinoan snowball Earth was a hard or a soft snowball Earth, which so far is unresolved, e.g.[22,23]. (4) The ocean could have gained large amounts of alkalinity (TA) through sub-snowball volcanism and weathering[24], but the exact concentration at the start of the deglaciation is unknown (see ref. [25] for a detailed description of the concept of alkalinity). We account for these uncertainties in the initial conditions by having a set of ICON-ESM simulations that cover a variety of possible scenarios (Fig. 1a). As the numerical stability and computational cost of ICON-ESM only allow for a limited range of simulations, we extend our analysis by box model calculations that cover a wider spectrum of possible initial conditions.

By accounting for the above-named uncertainties and resolving fast carbon cycle processes in the snowball Earth aftermath, we show that a slow, long-term decline of the supergreenhouse climate is indeed not the only possible scenario. Other pathways range from a rapid decline of the atmospheric $CO_2$ concentration to a supergreenhouse climate that intensifies during the first several thousand years (Fig. 1b). The large range of scenarios is a consequence of how the various carbon cycle processes by which the ocean impacts the atmospheric $CO_2$ concentration depend on the uncertain post-snowball Earth conditions.

### How the ocean modulates atmospheric $CO_2$

The deglaciation of large ice sheets and the massive warming in the snowball Earth aftermath change the ocean's temperature, salinity, chemical composition and volume (see Figs. S1 and S2 for time series from the model simulations). Additionally, the new climatic conditions lead to a transition from a largely homogeneous sub-snowball ocean[26] to a state with strong vertical and lateral variations[10]. In Fig. 2 we depict five carbon cycle processes that lead to a major exchange of carbon between the atmosphere and the ocean in response to these transformations. The individual processes are quantified in isolation from each other, for instance by running simulations with and without biological activity to determine the impact of biological carbon uptake

on the evolution of atmospheric $CO_2$. For quantifying the processes that depend on the chemistry of the ocean, we conduct additional box model calculations to cover a two-dimensional uncertainty space that is spanned by the uncertainties in the atmospheric $CO_2$ concentration ($10^4$–$10^5$ ppm) and the ocean alkalinity (1–60 mol m$^{-3}$) at the start of the deglaciation. Simulations with ICON-ESM are used to verify the numbers found with the box model calculations, to study the carbon cycle processes that are not easily quantifiable with a box model and to study the overall response of a system that integrates all processes. Generally, all numbers that are produced by our models should be seen as first-order approximations only. The goal of this paper is not to give a precise quantification of the carbon cycle processes that are important in the snowball Earth aftermath. Instead, we aim to establish a basic understanding of the different carbon cycle processes and their influence on the evolution of the supergreenhouse climate under the given uncertainties. All processes shown in Fig. 2 are part of the ocean's carbon cycle, and we discuss them one by one in the following sections.

### Meltwater inflow dilutes the ocean chemistry

The dilution of the sub-snowball brine by the inflow of meltwater reduces, among all other tracer concentrations, the concentration of dissolved inorganic carbon (DIC) and hence the $CO_2$ partial pressure (p$CO_2$) in the ocean, which will lead to a flux of $CO_2$ from the atmosphere into the ocean. We first quantify this effect with an idealised calculation that builds on the uniformity of the sub-snowball ocean. In this case, the ocean-atmosphere system can be represented by just two boxes and the effect of the inflow of meltwater can be calculated using simple carbonate chemistry. We consider a scenario where pure freshwater with a volume of 1000 m of sea-level equivalent is instantaneously mixed into a cold ocean previously in equilibrium with atmospheric $CO_2$. This reduces all tracer concentrations in the ocean by about 30%, leading to a lower p$CO_2$ in all cases. In the calculations, $CO_2$ is then removed from the atmosphere and added to the ocean in the form of DIC until a new equilibrium is found between the ocean surface p$CO_2$ and atmospheric $CO_2$. Over almost the entire two-dimensional uncertainty space defined by atmospheric $CO_2$ and TA, the atmospheric $CO_2$ concentration is reduced by 10–15%, with larger reductions only for very low initial atmospheric $CO_2$ (Fig. S3). For reference, we also quantify the dilution effect in a more realistic setup by comparing two of our ICON-ESM simulations (Exp. 3.1 and Exp. 4), which only differ by a strong dilution effect that is comparable to the idealised calculations described above. At the end of the simulations, the atmospheric $CO_2$ concentration is about 11.5% smaller in the experiment including the strong dilution effect. This reduction is comparable to the reduction derived with the idealised box model

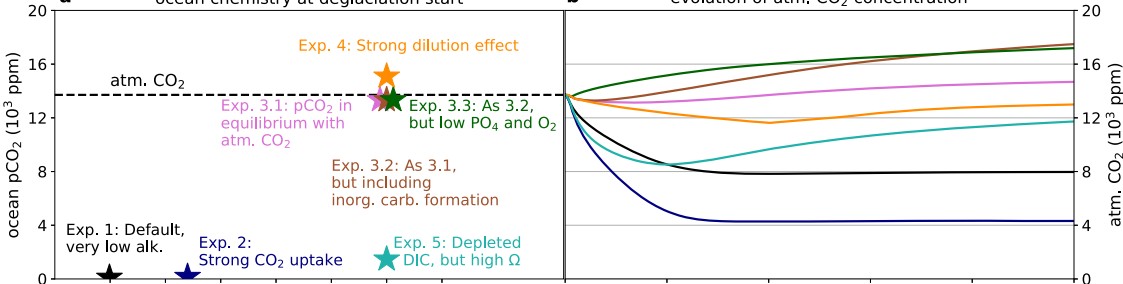

**Fig. 1 | The evolution of atmospheric $CO_2$ in ICON-ESM simulations with varying initial conditions. a** The initial alkalinity and partial pressure of carbon dioxide (p$CO_2$) in the ocean for the different ICON-ESM simulations. The initial atmospheric

$CO_2$ concentration (atm. $CO_2$) was around 13,700 ppm in all simulations, indicated by the dashed line. **b** Evolution of atm. $CO_2$ in the ICON-ESM simulations. More details on the experiment settings are found in the Method section.

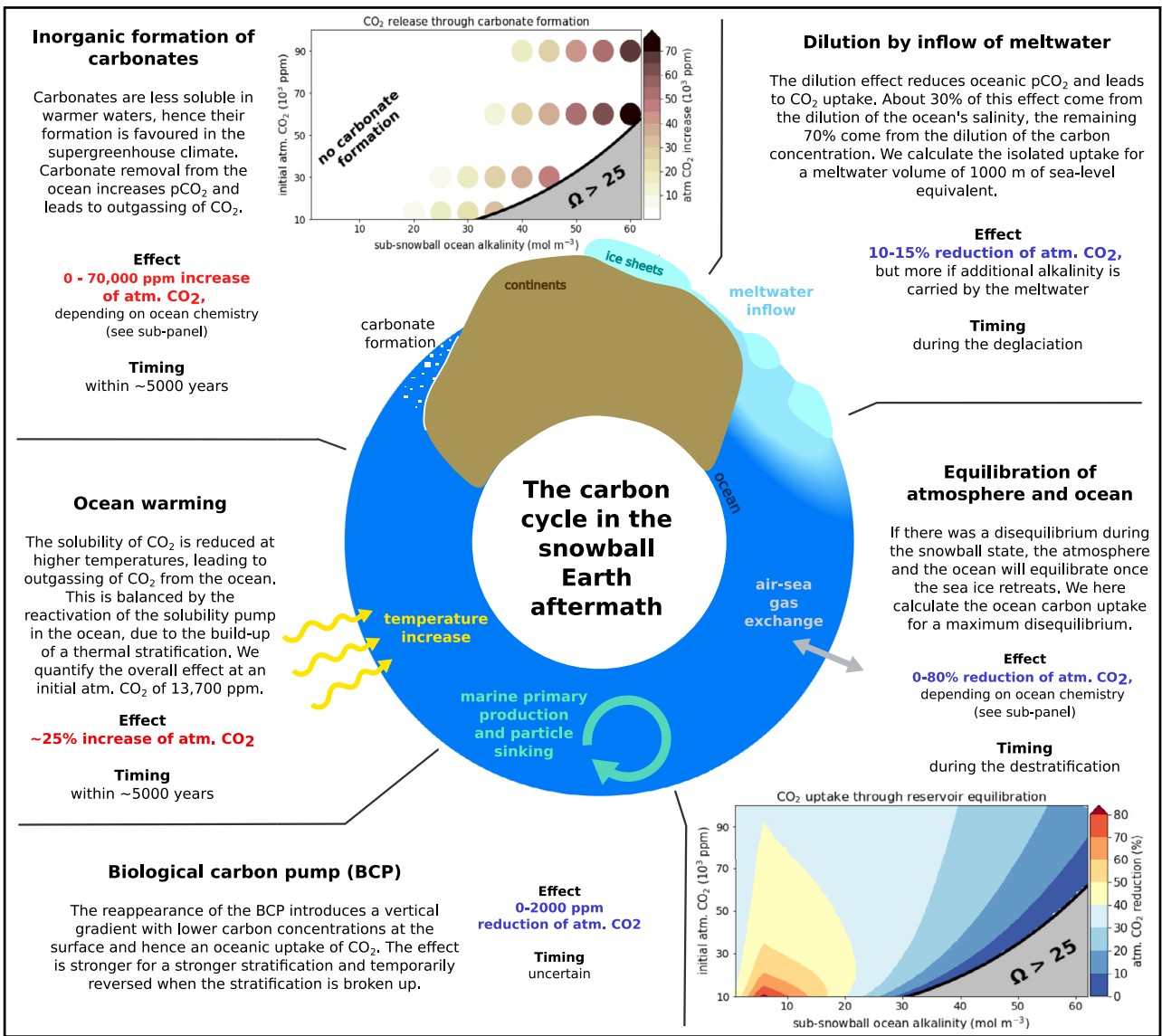

**Fig. 2 | Fast carbon cycle processes in the snowball Earth aftermath.** The figure summarises the five processes that lead to air-sea $CO_2$ exchange in the aftermath of a snowball Earth and were quantified in this study. The effect of the processes on atmospheric $CO_2$ (red text represents an increase and blue text represents a reduction of the atmospheric $CO_2$ concentration) and a rough estimate of the process time scale is given. The quantification is based on the ICON-ESM simulations and additional box model calculations.

calculation, which predicts a 13.5% reduction of atmospheric $CO_2$ for the conditions used in ICON-ESM.

The $CO_2$ uptake potential of the dilution effect depends strongly on the volume as well as the chemistry of the inflowing freshwater. We have assumed that the deglacial waters are devoid of any biogeochemical tracers. However, some DIC and TA are normally also present in sea ice, often with a ratio of TA:DIC that is larger than one[27,28], and further alkalinity could come with the meltwater flushes from the continents[29]. Therefore, DIC in the ocean could easily be diluted stronger than TA, which would lower pCO$_2$ even further, causing an even stronger dilution effect than predicted here.

## Equilibration of atmosphere and ocean

The reservoir equilibration effect is a process that accounts for the uncertainty about the ocean's carbon reservoir during the snowball state. Did the global sea-ice cover allow for an efficient exchange of $CO_2$ between the atmosphere and the ocean? If yes, then the two reservoirs were already in equilibrium at the start of the deglaciation and the magnitude of this process is zero, and air-sea $CO_2$

exchange is then only a consequence of other processes like the dilution effect or ocean warming, which are quantified separately. If not, then it is likely that the ocean's pCO$_2$ was substantially lower than that of the atmosphere, and the ocean takes up $CO_2$ from the atmosphere as the sea ice retreats. Because it is unclear how much carbon there was in the ocean, we assume here a minimal oceanic DIC concentration, which gives us the maximum magnitude of this effect. However, since an ocean without any carbon is unrealistic and would suck up all atmospheric $CO_2$ within a few hundred years from the start of the deglaciation, we add two requirements that the ocean carbon reservoir has to fulfil: (1) The ocean surface pCO$_2$, directly below the sea ice, is still >100 ppm, which is roughly the atmospheric $CO_2$ concentration needed for snowball initiation in our model. (2) The calcium carbonate saturation state $\Omega$ must be lower than 25, as otherwise carbonates would be formed already during the snowball state, and conditions for a given set of DIC and TA are unsustainable. We choose a relatively high threshold value of $\Omega = 25$ to reflect the difficulty to initiate inorganic carbonate precipitation in a cold sub-snowball ocean.

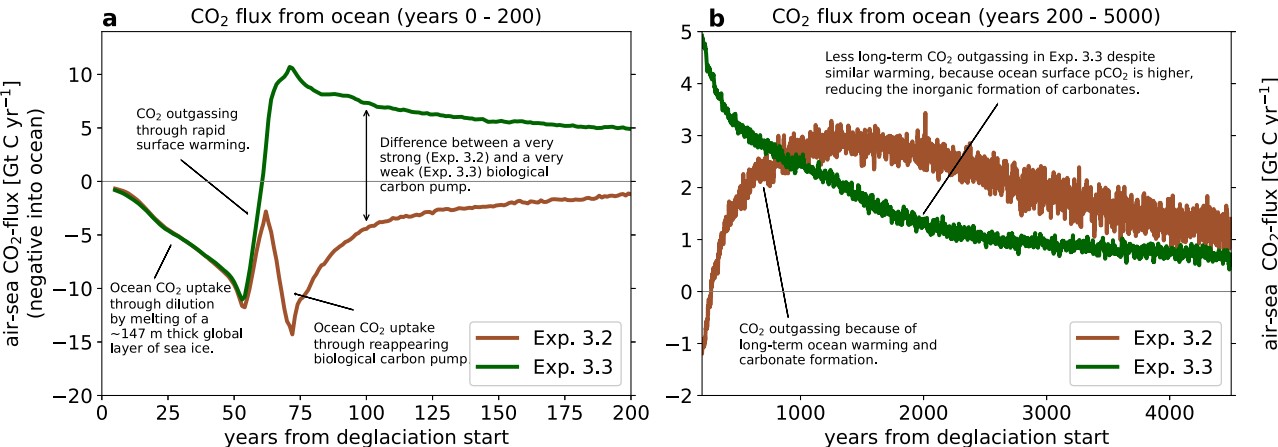

**Fig. 3 | Air-sea CO₂ exchange in experiments with high (Exp. 3.2) and low (Exp. 3.3) oceanic phosphate concentration.** The panels show the 10 year running mean air-sea CO₂ exchange (negative into the ocean) for the years 0–200 in (**a**) and 200–5000 in (**b**). The major drivers of each excursion are annotated with text in the figure, but it should be noted that their timing and magnitude are highly dependent on the setting of the simulation and the simplifications of the model. Hence, despite showing actual simulation results, this figure should rather be seen as an illustration of the effect of the different processes on the air-sea CO₂ exchange.

We again quantify this process with box model calculations, where we assume a scenario in which the sea-ice cover is suddenly permeable for the exchange of CO₂. The relative reduction of atmospheric CO₂ due to the reservoir equilibration effect is shown in the lower-right sub-panel of Fig. 2. At very low TA (<3 mol m⁻³), the atmospheric carbon reservoir is of a comparable size or even larger than the ocean carbon reservoir. Hence, already small relative reductions in atmospheric CO₂ increase the ocean pCO₂ substantially and lead to a new equilibrium. At high TA (>10 − 15 mol m⁻³), the large carbonate saturation states do not allow for a significant disequilibrium between the two reservoirs, and the relative reduction in atmospheric CO₂ through the reservoir equilibration effect is small as well. However, there is a maximum of the relative reduction of atmospheric CO₂ centred at very low initial atmospheric CO₂ and intermediate to low TA concentrations (3–10 mol m⁻³), where the reservoir equilibration effect can reduce the atmospheric CO₂ concentration by around 80%. Here, TA is large enough to accommodate a substantial flux of carbon into the ocean, and the carbonate saturation states are still low, so that a large difference between the ocean's pCO₂ and atmospheric CO₂ can be sustained. These calculations show that the reservoir equilibration effect could end the supergreenhouse climate already during the deglaciation, for specific combinations of TA and initial atmospheric CO₂.

Three ICON-ESM simulations (Exp. 1, Exp. 2, Exp. 5) are designed with a depleted ocean DIC, in order to test the possibility of a significant reduction of atmospheric CO₂. Many more processes are affecting the carbon cycle in these simulations, but the large reductions in atmospheric CO₂ predicted by the box model mean that the carbon cycle evolution is plausibly dominated by the reservoir equilibration effect. The predicted and simulated relative reductions in atmospheric CO₂ are, respectively, 48% and 42% for Exp. 1, 72% and 68% for Exp. 2 and 54% and 14% for Exp. 5. The respective values are very similar in Exp. 1 and Exp. 2, showing that the idealised calculations are approximately correct and that the reservoir equilibration effect indeed dominates the air-sea CO₂ exchange. The fact that the simulated reduction is slightly smaller than the predicted reduction indicates that the sum of all other carbon cycle processes leads to an overall outgassing in these simulations. In contrast, the simulated reduction in Exp. 5 is significantly smaller than the predicted oceanic uptake of CO₂. In that experiment, an additional process plays a major role and counteracts the CO₂ uptake by the reservoir equilibration effect. This process is the formation of carbonates from the ocean, which is quantified separately later. It should be noted again that the

ICON-ESM simulations Exp. 1, Exp. 2 and Exp. 5 and the idealised calculations of the reservoir equilibration effect are designed in order to achieve a maximum oceanic uptake of CO₂. The possible reductions in atmospheric CO₂ that we present here should therefore be seen as a maximum value, and the reservoir equilibration effect could also play a smaller role if more carbon resided in the ocean at the start of the deglaciation.

## Reactivation of the biological carbon pump

Phytoplankton and cyanobacteria will resume a stronger activity at some point in the snowball Earth aftermath, as the sea ice wanes and surface temperatures increase. This means that the biological carbon pump, which effectively moves carbon from the surface to the deeper ocean, will reactivate and lead to an oceanic uptake of CO₂ from the atmosphere. In most of our ICON-ESM simulations, primary production quickly ramps up to very high values, because oceanic phosphate concentrations are high. By comparing one of the simulations with strong biological activity to an otherwise identical test simulation with suppressed primary production, we find that the biological carbon pump reduces atmospheric CO₂ by around 2000 ppm in these simulations (Fig. S4). However, if the oceanic phosphate inventory is much lower at the start of the deglaciation (Exp. 3.3), this strongly limits biological activity and thereby the effectiveness of the biological carbon pump. In Exp. 3.3, the combined primary production of phytoplankton and cyanobacteria is reduced by 95%, when compared to Exp. 3.2 with higher phosphate concentration. Figure 3 shows the air-sea CO₂ flux in these two simulations, which start to differ around 50 years after the start of the deglaciation, when the area of ice-free ocean becomes considerable and biological activity starts to resume. As the biological carbon pump is much weaker at low phosphate concentrations, there is less transport of carbon into the deep ocean and hence a higher surface pCO2. This initially leads to the much stronger outgassing of CO₂ during the first ~1000 years from the start of the deglaciation. However, this also reduces the process of rapid carbonate formation, which is discussed separately later, leading to less long-term outgassing of CO₂. The biological carbon pump and the biology-induced part of carbonate formation roughly balance each other out, and the atmospheric CO₂ concentration is almost identical at the end of the two simulations. This means that in scenarios where the ocean carbonate saturation state is generally high enough to allow for inorganic formation of carbonates, there is a direct positive relationship between biological activity and that part of carbonate formation that is induced by biological activity. The overall effect of the

biological carbon pump therefore only leads to a substantial reduction of atmospheric $CO_2$, if the carbonate saturation state is low enough that the surface $pCO_2$ reduction does not induce inorganic carbonate formation.

In addition to the uncertain magnitude of the effect of the biological carbon pump, also its timing is uncertain. Primary production reappears rapidly after the snowball Earth in our model, because it is implemented in HAMOCC to represent the modern primary producers of the ocean[30]. However, it is questionable whether those biota that adapted to a snowball Earth state could indeed thrive and spread quickly in the hot snowball Earth aftermath. It is conceivable that there would be at least a delay in the reappearance of the biological carbon pump, while the biology adapts to the changed climatic boundary conditions. The temporal uncertainty of the effect of the biological carbon pump is increased further by the changing ocean circulation. The meridional overturning circulation (MOC) recovers quickly in most of our ICON-ESM simulations (Fig. S1d), because the amount of meltwater is comparably small and because the dynamics of the ocean circulation break up the stratification quickly[10]. The MOC is suppressed for a longer time only in the simulation with a sustained freshwater influx (Exp. 4). The recovery of the MOC, which is best visible around year 3000 in Exp. 4, then triggers a period of increased outgassing, because the upwelling of carbon-rich deep waters strengthens and the effectiveness of the biological carbon pump is reduced due to the weakening stratification. Overall, we expect that the reappearance of the biological carbon pump reduces atmospheric $CO_2$ by up to 2000 ppm at most, if a high abundance of nutrients fuels extensive primary production and the carbonate saturation state is low enough so that there is no counter-balancing biology-induced carbonate formation. However, the timing of this effect is uncertain and the actual magnitude could also be much smaller if nutrient availability is reduced.

## Ocean warming drives solubility changes

The solubility of $CO_2$ in seawater decreases with rising temperatures. During the snowball state, ocean temperatures are close to the freezing point, which corresponds to -1.8°C in ICON-ESM. Global mean sea-surface temperatures then increase to values around 20°C within just a few hundred years from the start of the deglaciation in our simulations, and they may reach 25-30°C in the long term (Fig. S1b). However, it is not practical to calculate the coupled atmospheric $CO_2$ change for this mechanism in a box model calculation, because of the large spatial variations of temperature within the ocean. Specifically, a thermal stratification develops with higher temperatures at the surface than there are in the deep ocean. This means that the carbon solubility pump (e.g. [31]), which was not active in the well-mixed snowball ocean, comes into play again and reduces the warming-induced $CO_2$ release substantially. Because of this entanglement, we can only quantify the combined effect of the reduced solubility and the reintroduced thermal stratification in our ICON-ESM simulations, where also other processes are active. We estimate that the warming of the ocean increases the atmospheric $CO_2$ concentration by about 25% in our ICON-ESM simulations. This is derived from Exp. 3.1, in which the reservoir equilibration effect is zero, the dilution effect is weak, the formation of carbonates is inactive and the biological carbon pump is crudely estimated to reduce atmospheric $CO_2$ by 2000 ppm. It can further be seen from Fig. 3 that a large part of the warming-induced outgassing of $CO_2$ occurs during the first several hundred years, but it is counter-acted by both the dilution effect and the reappearing biological activity. However, on a longer time scale of a thousand years or more, these other processes weaken quicker than the warming effect, which continues to lead to some outgassing of $CO_2$. If the atmospheric $CO_2$ concentration was even higher, the warming would be even stronger, leading to even more outgassing of $CO_2$, but the relative increase in atmospheric $CO_2$ would be smaller.

## Carbonate formation driven by warming and biological activity

Carbonate forms inorganically in seawater if the saturation state $\Omega$ is above a value of critical supersaturation[32–34]. This saturation state is highly dependent on the amount of carbon and alkalinity in the ocean, but also on the ocean's temperature and salinity[35]. During the snowball Earth, the ocean is very cold, the salinity is high, and the absence of biologically produced particles that could act as condensation nuclei for inorganic formation of carbonates suggest that large amounts of alkalinity could be stored in the snowball ocean at the start of the deglaciation. This alkalinity was possibly provided by shallow-ridge volcanism during the snowball Earth period[24]. When the snowball Earth deglaciates, the saturation state of carbonates initially decreases due to the dilution of tracers during the inflow of meltwater, but subsequently increases again due to the strong warming of the ocean. Additionally, biological activity reduces $pCO_2$ in the surface ocean, further increasing the carbonate saturation state. If the saturation state surpasses the supercritical threshold value, carbonates form inorganically and then sink downward into the sediment. This changes the ocean's chemistry and effectively increases $pCO_2$, leading to outgassing of $CO_2$ from the ocean. Therefore, ocean warming and the reappearing biological activity not only have a direct impact on the carbon cycle, as described in the sections above, but also an indirect impact, because they facilitate rapid carbonate formation by reducing carbonate solubility. Interestingly, only the warming-induced part leads to an overall outgassing of $CO_2$, because the biology-induced part by definition only occurs if the biological carbon pump has previously reduced $pCO_2$ in the surface ocean and led to oceanic $CO_2$ uptake before. It is further important to distinguish the rapid warming-induced formation of carbonates from the more long-term carbonate formation through the import of alkalinity by rivers and weathering. In the latter case, alkalinity is first added to the ocean before it is removed again, which has a net effect of zero on atmospheric $CO_2$, while the processes we discuss here involve a net removal of alkalinity from the ocean and hence the outgassing of $CO_2$ into the atmosphere.

The process of inorganic precipitation of carbonates from the water column is implemented in the ICON-ESM simulations Exp. 2 (no effect), Exp. 3.2, Exp. 3.3 and Exp. 5. As Exp. 3.2 is identical to Exp. 3.1, except for the activated inorganic carbonate formation, the difference in the atmospheric $CO_2$ concentration between the two experiments gives us the isolated effect of carbonate formation on the carbon cycle for the specific set of conditions applied in this simulation: atmospheric $CO_2$ is ~ 2800 ppm higher if carbonate formation is activated. However, the effect is highly dependent on the ocean's chemistry, which is why we also conduct a set of carbonate chemistry model calculations that extend the ICON-ESM simulations to a larger range of ocean chemistry conditions and initial atmospheric $CO_2$ concentrations. These calculations are designed to reconstruct the amount of carbonate that could have possibly precipitated from supersaturated ocean waters, and they are described in more detail in the Methods section. We note that, due to the large range of uncertainties and model simplifications, the numbers that are produced by the reconstruction should only be seen as a first-order approximation. The results are shown in the sub-panel in the upper left corner of Fig. 2. We find that, depending on the ocean chemistry at the start of the deglaciation, this process could increase the atmospheric $CO_2$ concentration by as much as 70,000 ppm in the case of very high alkalinity and strong warming, or not come into play at all.

## On the origin of Marinoan cap dolostones

In the aftermath of the Marinoan snowball Earth, large formations of carbonate rocks were created. The Marinoan cap dolostones thereby show several signs of rapid accumulation during a period of sea-level rise[3,36–38] and possibly under the influence of a deglacial surface freshwater layer[39–42]. This indicates a deposition time scale of between a few thousand years and a maximum upper bound of 60 ky, which was

derived from glacio-isostatic adjustment calculations[3,43]. At the same time, however, the dolostones include magnetic reversals that hint at much longer deposition time scales of $10^5$–$10^6$ years[44,45]. This discrepancy has been discussed in ref. 3 and might be resolved by a later inception of the Earth's inner core and a potentially increased frequency of magnetic reversals for a missing solid inner core[46]. After all, a convincing mechanism for cap dolostone formation is missing for the case that Marinoan cap dolostones indeed accumulated rapidly. The carbonate formation from pre-existing ocean alkalinity described in this study could be such a mechanism, since it is driven by the fast warming of the surface ocean and a potentially rapid reappearance of biological activity.

Indeed, in our simulations, most carbonates are formed in the tropics and subtropics and the deposition mostly occurs during the first 2000 years after the start of the deglaciation (Fig. 4). However, the calculated time scale of deposition could be slowed down by one or two thousand years through a better representation of meltwater inflow during the deglaciation. Furthermore, these calculations are based on a simulation with a very strong and rapidly evolving biological carbon pump, which also contributes to the high rates. At the same time, the continental meltwater flushes would likely carry additional alkalinity[29], on the one hand, shifting the location of carbonate precipitation more towards coastal regions, on the other hand, increasing the available alkalinity for carbonate accumulation and further prolonging the time scale of deposition. The carbonate deposition inferred from this study is large enough to explain massive carbonate formations deposited on a short time scale, as long as the initial ocean alkalinity is high enough. But due to the lack of resolution and knowledge about regional topography, our model can reconstruct neither the exact distribution of Marinoan cap dolostones nor its precise time scale. Nevertheless, the overall accumulation mostly in low latitudes and the rapidness of deposition lead us to the conclusion that the carbonate formation from pre-existing alkalinity in the sub-snowball ocean could have been a major contributor to the creation of the Marinoan cap dolostones, if these rocks were indeed deposited within a few thousand years after the start of the deglaciation.

## Discussion

We have identified and quantified five fast carbon cycle processes that influence the atmospheric $CO_2$ concentration within the first few thousand years after the termination of a snowball Earth. On the one hand, there are three predictable processes which certainly occurred, but their effects on atmospheric $CO_2$, despite being substantial, do not have the potential to lead to a change in the pathway of the post-snowball Earth climatic evolution. These processes are the inflow of meltwater into the ocean, the reactivation of the biological carbon pump and the general warming of the ocean. It does not become clear from our model simulations whether the combined effect of these three processes leads to an overall outgassing or an uptake of $CO_2$ by the ocean, because the processes all have uncertainties in their magnitude and timing and will in part cancel each other out. On the other hand, there are two processes with large individual magnitudes, which could on their own shape the climatic evolution of the snowball Earth aftermath: the reservoir equilibration effect and the formation of carbonates from a high-alkalinity ocean, driven by the rapid warming and reviving biological activity. These processes have opposing effects on atmospheric $CO_2$, and both are highly dependent on the carbonate chemistry of the ocean and the atmospheric $CO_2$ concentration at the start of the deglaciation. The set of possible starting conditions can be illustrated by means of a two-dimensional space defined by the uncertainty ranges of the atmospheric $CO_2$ concentration and ocean alkalinity (Fig. 5). Substantial rapid formation of carbonates and hence outgassing of $CO_2$ would only occur on the right side of this plot, approximately at ocean alkalinities above 10-20 mol m$^{-3}$ (brown shading in Fig. 5). By contrast, a strong reduction of atmospheric $CO_2$ by the reservoir equilibration effect would be most effective at low ocean alkalinities, under the additional premise that the ocean was depleted in carbon at the start of the deglaciation. Together, this explains why the evolution of the supergreenhouse climate can differ substantially depending on the conditions at the start of the deglaciation.

The geological record might help to constrain the range of possible scenarios for the evolution of the supergreenhouse climate. If Marinoan cap dolostones indeed formed within a few thousand years after the start of the deglaciation[3,36–42], low ocean alkalinity values, especially at high atmospheric $CO_2$ concentrations, are unlikely, because the ocean would be too acidic to allow for a rapid deposition of carbonates. Additionally, some geochemical proxy data indicate maximum $CO_2$ concentrations of about $10^5$ ppm and maybe substantially lower concentrations just shortly after the snowball Earth period[4,47]. If we assume these proxies to be valid, this excludes scenarios with very high ocean alkalinities and atmospheric $CO_2$ concentrations, as in those scenarios the rapid formation of carbonates would increase the atmospheric $CO_2$ concentration by too much.

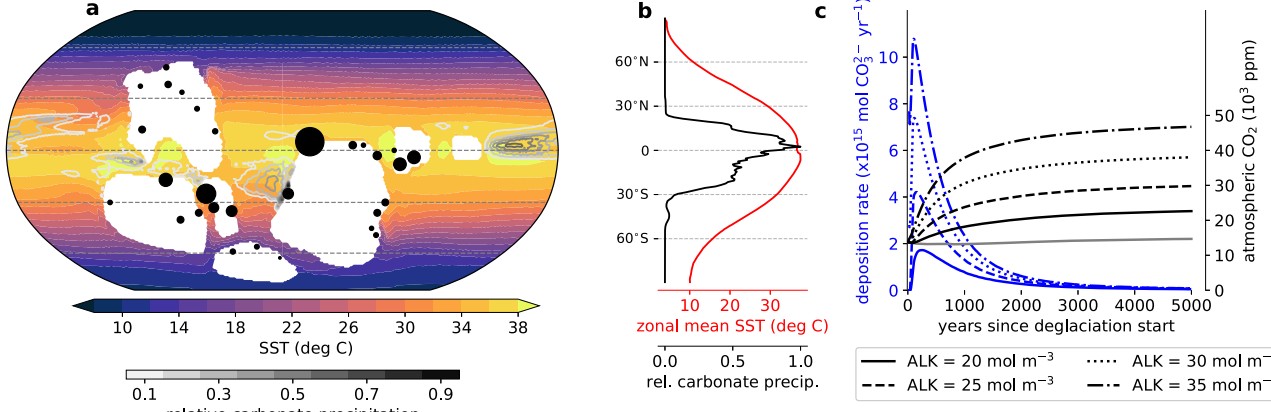

**Fig. 4 | Patterns of inorganic carbonate formation induced by ocean warming and biological activity after a snowball Earth. a** The mean sea-surface temperature (SST) averaged over the full 5000 years from the start of the deglaciation, overlain by contour lines of relative carbonate formation as simulated with the ICON-ESM simulation Exp. 3.2. The black dots indicate the location of actual Marinoan cap dolostones (data from ref. 36), and the area of the dots is proportional to the thickness of the cap dolostone formation. **b** Zonal averages of the data in (**a**). **c** Annual carbonate formation rates (blue lines) for the reconstructions of the carbonate chemistry model with ocean $pCO_2$ = 13, 200 ppm at the start of the deglaciation. The black lines show the rise in atmospheric $CO_2$ corresponding to the carbonate formation in the reconstructions. The grey line represents the evolution of atmospheric $CO_2$ when no carbonate is formed.

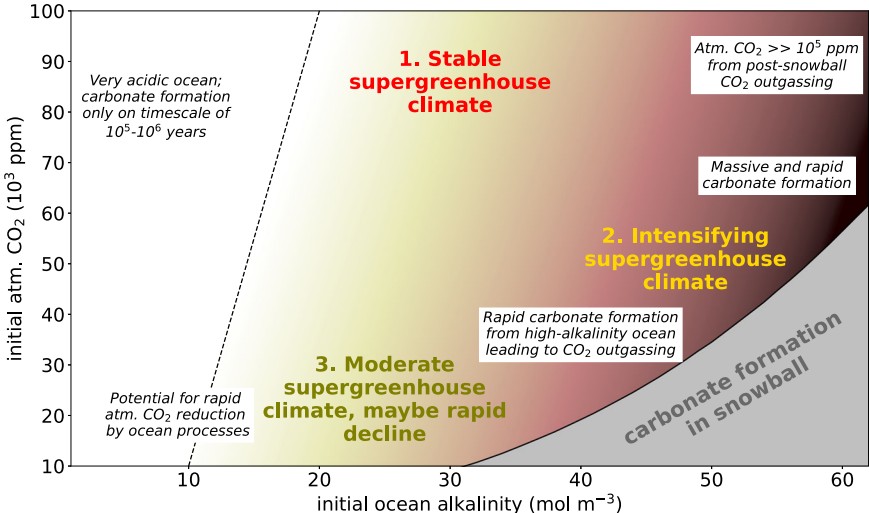

**Fig. 5 | The evolution of the supergreenhouse climate after a snowball Earth, depending on the initial atmospheric CO₂ concentration and ocean alkalinity.** Based on the carbon cycle processes quantified in this work, the coloured labels indicate the expected climate scenario at a given set of initial conditions. The smaller text boxes point to specific characteristics of the carbon cycle during the snowball Earth aftermath. The brown shading illustrates where rapid carbonate formation through ocean warming and biological activity could occur (upper sub-panel from Fig. 2). The grey area in the lower right corner represents combinations of atmospheric CO₂ and alkalinity that are unsustainable during a snowball Earth due to unrealistically high carbonate saturation states. To the left of the dashed line, the acidic ocean only allows for carbonate formation on very long time scales. The position of the line should be seen as a rough indication.

Taking these considerations into account, we can describe the evolution of the supergreenhouse climate during the first several thousand years after a snowball Earth through three plausible scenarios (Fig. 5): 1. The atmospheric $CO_2$ concentration persists at a high initial value, because no process leads to a major exchange of $CO_2$ with the ocean. 2. The supergreenhouse climate intensifies, if the rapid formation of carbonates from a high-alkalinity ocean dominates the carbon cycle and leads to massive oceanic outgassing of $CO_2$. 3. The supergreenhouse climate is moderate and maybe even declines rapidly when the initial atmospheric $CO_2$ concentration is on the lower end of the spectrum and carbon cycle processes leading to oceanic uptake of $CO_2$ play a major role. Our results imply that pathways in which the supergreenhouse climate was intense and only declined through long-term continental weathering (scenarios 1 and 2) are plausible also after accounting for short-term ocean carbon cycle dynamics. However, we highlight that a more moderate and possibly short-lived supergreenhouse climate is an equally plausible scenario. In fact, while all scenarios are in line with the possibility of a rapid accumulation of carbonates, scenario 3 could also explain lower atmospheric $CO_2$ concentrations shortly after the deglaciation and imposes less severe conditions for early forms of life in the snowball Earth aftermath. We therefore conclude that it is plausible that the supergreenhouse climate was moderate and possibly short-lived and that Marinoan cap dolostones could have indeed formed in just a few thousand years after the start of the deglaciation, driven by the rapid warming of the ocean and the reappearance of biological activity.

## Methods
### ICON-ESM model description
The icosahedral nonhydrostatic model ICON-ESM is a recently developed Earth system model with an unstructured grid and quasi-uniform grid cell areas[17]. It couples the atmosphere general circulation model ICON-A[48] with the ocean general circulation model ICON-O[49] and the ocean biogeochemistry model HAMOCC[17,30]. It furthermore includes an interactive carbon cycle, meaning that carbon is handled as a mass-conserving, prognostic tracer in all components. We here use ICON-ESM in a coarse resolution setup that is adapted to the boundary conditions of the Marinoan snowball Earth. The setup of the two physical components (atmosphere and ocean) is largely identical to

the model version that was used and described in more detail in[10], including now an improved representation of sea-ice dynamics[50]. The HAMOCC model has been successfully used in previous paleo-modelling efforts such as for the Paleocene-Eocene Thermal Maximum[51,52] or the last deglaciation[53]. Therefore, we here only present a description of the specifics of the ocean biogeochemistry component that are important for this study and refer the reader to the cited literature for more information.

The HAMOCC model describes the biogeochemical dynamics of at least 17 prognostic state variables that are advected by the circulation calculated in the ocean model. These tracers include total alkalinity (TA) and dissolved inorganic carbon (DIC), which are sufficient to calculate the carbonate chemistry, as well as tracers for primary producers, the main nutrients, dissolved gases and further biogeochemical components[30]. As the model is run with an interactive carbon cycle, HAMOCC receives information on the concentration of $CO_2$ in the overlying atmospheric grid cell at every ocean time step (60 min) and calculates the surface flux of $CO_2$ based on the difference to the $pCO_2$ in the ocean surface layer. We tuned HAMOCC for a good representation of the modern biogeochemistry in a setup with present-day continents and then use the calibrated parameter settings in the Marinoan setup.

Some modifications of the biogeochemical implementations are made in order to adapt the model for an application to the Neoproterozoic conditions: (1) Shell production by phytoplankton was turned off, because both silicate shell producers and marine calcifiers had not yet developed in the late Neoproterozoic[54]. (2) The standard implementation of phytoplankton growth uses an exponential dependence on temperature in the default HAMOCC setup. This would lead to unrealistically high growth rates during the supergreenhouse climate. Therefore, we modified the dependence to let the growth rate of phytoplankton decrease at temperatures above 35 °C. Cyanobacteria growth is only limited by temperatures below 25 °C. (3) Iron input to the ocean, in the form of dust deposition at the surface, is modified to a high value that is constant in time and globally uniform, only reducing northwards in the ocean-dominated latitudes > 50°N. This accounts for the high rates of dust production on the bare Neoproterozoic continents[55]. (4) The atmospheric concentration of oxygen is assumed to be 50% of the present atmospheric level and is constant in time. The

real concentration after the Marinoan snowball Earth is uncertain, but it is known that atmospheric oxygen increased to modern-like values during the Neoproterozoic Oxidation Event, in close temporal relation with the Cryogenian glaciations[56–58]. (5) Calcium is added to the list of state variables in order to be able to simulate the inorganic formation of carbonates in some of the model experiments.

## Carbonate chemistry

The ICON-ESM simulations that include the formation of carbonates from supersaturated waters - also called inorganic carbonate precipitation - calculate the carbonate formation rate as

$$R = \begin{cases} k(\Omega_{CaCO_3} - \Omega_{thresh})^3 & \text{if } \Omega_{CaCO_3} > \Omega_{thresh} \\ 0 & \text{otherwise,} \end{cases} \quad (1)$$

following refs. 33,59, where $k$ is a rate constant and $\Omega_{thresh} = 15$ is the threshold saturation above which the inorganic carbonate precipitation starts. We use the saturation state of calcium carbonate, $\Omega_{CaCO_3}$, as a proxy for the inorganic precipitation of $CaMg(CO_3)_2$, because of the large uncertainties regarding the formation of dolomite[59,60]. We further assume that any TA concentration that is substantially higher than the modern concentration was added in the form of $Ca^{2+}$ or $Mg^{2+}$ and that the seawater ratio of $Mg^{2+}$ to $Ca^{2+}$ is 3:1. We do not account for any further input of these elements, which would be expected from intense continental weathering[8,9,29]. As we assume that all carbonate precipitates in the form of dolomite, only one mole of $Ca^{2+}$ is removed for every two moles of carbonate, and we only track $Ca^{2+}$ explicitly in the model for simplicity. These assumptions reflect the large uncertainties in carbonate precipitation dynamics and the seawater composition during the aftermath of the Marinoan snowball Earth. However, the results presented in this study are mostly of a qualitative nature, and different assumptions would only shift a certain scenario to a different location within the spread of uncertainties related to the oceans' chemical conditions of that time.

The idealised carbonate chemistry calculations that extend the ICON-ESM simulations are based on a model that is mostly a translation of the carbonate chemistry implementation of HAMOCC[30] into an easy-to-use python code. The model then derives the chemical conditions based on two state variables of the carbonate system, TA and DIC, using information about temperature, salinity and pressure for calculating the dissociation constants. Solutions to this carbonate chemistry system are described comprehensively in the literature (e.g. ref. 61). Some information on how this carbonate chemistry model is applied can also be found in the description of the meltwater dilution and reservoir equilibration carbon cycle processes in the Result section. In the following, we describe how the calculations to reconstruct inorganic carbonate formation were designed. The reconstructions use the annual mean simulation output of Exp. 3.1, which has an ocean $pCO_2$ initially equilibrated with the atmospheric $CO_2$ concentration and a sub-snowball ocean alkalinity of 15 mol m$^{-3}$. The simulation itself did not include the process of inorganic precipitation of carbonates. The fields of TA and DIC are adapted by adding globally uniform concentrations. The reconstruction model works with a time step of 1 year. In each time step, $\Omega_{CaCO_3}$ is calculated for every grid cell of the surface ocean, and if $\Omega_{CaCO_3} > \Omega_{thresh}$, TA, DIC, $Ca^{2+}$ and $Mg^{2+}$ are removed with a ratio of 2:1:0.5:0.5 until the saturation state in this grid cell is below the threshold. As this process increases $pCO_2$, DIC is removed from the global surface ocean in order to reflect outgassing of $CO_2$. The $CO_2$ concentration of the atmosphere is increased accordingly until a new equilibrium is established. The increased atmospheric $CO_2$ concentration then feeds back to the temperature in the surface ocean, assuming a constant climate sensitivity of 4.9 K per doubling of $CO_2$[10]. The depleted local reservoirs in each surface grid cell are then resupplied on two timescales from a global surface ocean and a deep ocean reservoir. All parameterizations of the carbonate chemistry model were tuned for a good reconstruction of the inorganic precipitation dynamics simulated in Exp. 3.2. The reconstructions allow us to directly infer the amount of $CO_2$ that is outgassed as a consequence of the formation of carbonates.

## Simulation design

All our simulations with ICON-ESM incorporate the ocean biogeochemistry component and a prognostic carbon cycle at all times, and transitions in the climate are initiated by prescribing positive or negative emissions of $CO_2$ into the atmosphere. As the focus of this study is on the aftermath of a snowball Earth, the initiation and the snowball state itself are simplified. An overview of the prescribed emissions and the evolution of major climate variables throughout the snowball cycle is presented in Figs. S5 and S6. The simulation procedure starts with a stable control climate at an atmospheric $CO_2$ concentration of 1500 ppm, leading to a global mean temperature of 8.5 °C. This control climate has the same settings as the control climate in ref. 10, but a newer version of ICON-ESM is used. The model is then forced into a cold pre-snowball state at 112 ppm $CO_2$ by prescribing negative emissions of $CO_2$. The snowball state itself is triggered from the stable pre-snowball climate by another short period of prescribed negative emissions, where 400 GT C are removed over 200 years. After 300 years without carbon emissions to allow for some growth of the sea ice, carbon is added to the atmosphere at a very high rate of 100 GT C yr$^{-1}$ for another 300 years. This causes a rise of the atmospheric $CO_2$ concentration to 13,710 ppm, which is not enough to initiate deglaciation in our model. As higher $CO_2$ concentrations would cause instabilities in the atmosphere component of the ICON-ESM model, we trigger the deglaciation through a 100 year long reduction of the albedo of snow on ice to a value of 0.6 (default values are between 0.66 and 0.79 depending on the surface temperature). The onset of this reduction is what we refer to as the start of the deglaciation in this paper, and from that point on the model simulations are let run freely without prescribing any further emissions of $CO_2$.

At the start of the deglaciation, the ocean under the ice is well mixed with largely uniform values. We make use of this property by adapting the inventories of some of the biogeochemical tracers to create a set of additional simulations. In the control climate and during the snowball cycle, the biogeochemical inventories of the ocean are still similar to modern values (a remnant of the initialisation of the model). These inventories remain unchanged in the reference simulation Exp. 1, but are adapted in all other simulations, as described below. Among the additional experiments, only TA and DIC differ, and we use these differences to cover a range of possible scenarios for the post-snowball evolution of the climate and the carbon cycle. A detailed overview of the different ICON-ESM simulations is presented in the following.

## List of experiments

The initial chemical conditions of the ocean in our ICON-ESM simulations and the conditions after 5000 simulation years can be found in Tab. S1. Apart from the default simulation Exp. 1, all other simulations adopt the idea of a chemically evolved ocean, which results from the chemical alteration of the seawater during the snowball state, driven by hydrothermal and weathering activities[15,24,62]. In these simulations, the phosphate inventory is increased to ~5 times the modern concentration[63]. After all simulations were done, we learned that the conclusions of ref. 63 are very debated and that much lower phosphate concentrations are possibly more likely at least during the Sturtian snowball Earth[62,64]. Nevertheless, the phosphate concentration after the Marinoan snowball Earth is very uncertain, and with our approach of using a high phosphate concentration, we make sure that primary production is not limited by phosphate. Instead it is limited by nitrate in our model simulation, which is consistent with geological findings[65]. We additionally conduct a reference simulation with much lower

phosphate and oxygen concentrations, where biological activity will be much weaker and limited by phosphate. In all simulations the iron concentration is increased by a factor of five, while oxygen is reduced to anoxic conditions, in order to replicate largely reducing conditions during the snowball state[66]. The nitrate inventory is unchanged, because of sparse information and the fact that the nitrate inventory is not closed due to the processes of denitrification and N2-fixation by cyanobacteria[65]. In the following, we present a short description of the individual experiments and their motivation.

**Experiment 1.** This is the default simulation that represents a continuous simulation from a pre-snowball control climate through the snowball state into the supergreenhouse climate without any adaptions of the inventories. In contrast to most other simulations, in Exp. 1 the ocean is not chemically evolved and the tracer inventories are similar to modern values. As the snowball cycle itself is short, the model has no time for an equilibration of the atmosphere and ocean carbon reservoirs, so that the dissolved inorganic carbon (DIC) reservoir in the ocean is largely depleted. Due to the low sub-snowball total alkalinity (TA) of $TA = 2.47$ mol m$^{-3}$, the calcium carbonate saturation state is far below the threshold for inorganic carbonate precipitation in the snowball state, and the pH of the ocean drops very quickly during the deglaciation, as the ocean equilibrates with the high atmospheric $CO_2$ concentration (atmospheric $CO_2$). All other simulations, except for Exp. 4, are started from the end of the snowball cycle in this simulation, so that the experiments only differ from the start of the deglaciation onward.

**Experiment 2.** Here, we try to achieve a maximum reduction of the atmospheric $CO_2$ concentration through the reservoir equilibration effect. Therefore, TA is 6 mol m$^{-3}$ and DIC is reduced to a very low value of 5.05 mol m$^{-3}$, resulting in an ocean pCO$_2$ of 179 ppm at the start of the deglaciation. As the sub-snowball calcium carbonate saturation state is already quite high, the inorganic precipitation of carbonates is activated in this simulation. However, it does not play a significant role, because the saturation state drops quickly during the phase of reservoir equilibration.

**Experiment 3.1 and 3.2.** Exp. 3.1 is a reference simulation for many other experiments, and it provides the simulation output that is fed into the reconstructions with the carbonate chemistry model. It has a comparably high TA of 15 mol m$^{-3}$ and the DIC concentration is chosen to be 15.75 mol m$^{-3}$, so that the ocean pCO$_2$ is in equilibrium with atmospheric $CO_2$ at the start of the deglaciation (13,335 and 13,710 ppm respectively). Exp. 3.2 is identical to Exp. 3.1, except that Exp. 3.2 includes the inorganic precipitation of carbonates, while this process is deactivated in Exp. 3.1. Because the two experiments differ only in this process, it allows for an isolated quantification of the effect of inorganic carbonate precipitation on the evolution of the climate, which we use in order to tune our reconstructions with the carbonate chemistry model to produce a similar behaviour.

**Experiment 3.3.** In this experiment we test the impact of very low phosphate and oxygen concentrations on the evolution of the carbon cycle. While all other simulations, except for the default experiment Exp. 1, adopt the idea that the ocean had gained substantial amounts of phosphate during the snowball Earth, the assumption here is that phosphate concentrations remained very low in order to sustain a low atmospheric $O_2$ concentration. This simulation has the identical settings as Exp. 3.2, but we reduce the mean phosphate concentration down to 0.1 mmol m$^{-3}$ (from 10 mmol m$^{-3}$). Additionally, the initial oxygen concentration under the ice is reduced to 0.01 mmol m$^{-3}$ (from 1.0 mmol m$^{-3}$), which is below the threshold for aerobic remineralisation in the model. The atmospheric concentration of oxygen is set to 0.1 PAL (present atmospheric level), while it is 0.5 PAL in all other simulations.

**Experiment 4.** The main purpose of this simulation is to quantify the effect of dilution on the oceanic carbon uptake. Here, a total freshwater volume of 1000 m of sea-level equivalent was removed over a timescale of several thousand years during the transition from our control climate into a cold pre-snowball state. Hence, this simulation is based on a different snowball cycle experiment than the other simulations. The snowball period itself is designed similar to the representation in the default simulation Exp. 1, with a terminal mean sea-ice thickness of 149 m. Together with the start of the deglaciation, a globally distributed freshwater flux is implemented, adding every year freshwater with a volume of 0.5 m of sea level back to the ocean over a timescale of 2000 years. As this freshwater input suddenly stops in year 2000 from the deglaciation start, most time series of this simulation show a kink at this time (Figs. S1 and S2). The biogeochemical concentrations of the ocean at the start of the deglaciation are identical to those in Exp. 3.1, so that all differences to Exp. 3.1 are direct or indirect consequences of the dilution effect.

**Experiment 5.** Exp. 5 has the same TA as Exp. 3.1, but DIC is reduced to a very low value, which leads to a calcium carbonate saturation state of 24.3 at the start of the deglaciation. This simulation was designed in order to study the interplay of the competing effects of the oceanic uptake of $CO_2$ through the reservoir equilibration mechanism and the possible outgassing of $CO_2$ during the inorganic precipitation of carbonates. Inorganic precipitation of carbonates is implemented as described in Eq. (1), so that here the initial calcium carbonate saturation state is already larger than the threshold. This is motivated by the idea that during the snowball state there are no organic particles that could function as condensation nuclei, and higher saturation states than in an ice-free ocean are sustainable. Due to the melting of the sea ice and the equilibration with the high atmospheric $CO_2$, the oversaturation initially drops below the threshold value, but then increases again, as the ocean warms and waters with higher alkalinity are mixed to the surface.

## Data availability
The model output of ICON-ESM that was used to create the figures in this manuscript, as well as all scripts used for postprocessing the data and creating the figures are available in a permanent repository at the Climate and Environmental Retrieval and Archive (CERA) of the World Data Center for Climate (WDCC) (https://www.wdc-climate.de/ui/entry?acronym=DKRZ_LTA_033_ds00016). The repository also includes a detailed description of the simulation history and the required settings for reproducing the ICON-ESM simulations.

## Code availability
The analysis presented in this manuscript was conducted using Python scripts, which are available under the BSD-3-Clause license in a permanent repository at WDCC (https://www.wdc-climate.de/ui/entry?acronym=DKRZ_LTA_033_ds00016). The exact version of ICON-ESM that was used in this study for the Earth system model simulations is stored in a separate repository at WDCC (https://www.wdc-climate.de/ui/entry?acronym=DKRZ_LTA_033_ds00017). The source code of ICON-ESM is available under the "personal non-commercial research ICON license" (https://code.mpimet.mpg.de/attachments/download/20888/MPI-M-ICONLizenzvertragV2.6.pdf).

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

## Acknowledgements

We thank Bo Liu for the internal review of the manuscript. This work was supported by the Max Planck Society for the Advancement of Science and the International Max Planck Research School on Earth System Modelling. T.I. was supported by the European Union's Horizon 2020 research and innovation programme under grant agreement no. 101003536 (ESM2025 - Earth System Models for the Future). All model simulations and analyses were performed using resources of the German Climate Computing Center (DKRZ).

## Author contributions

T.I. and J.M. conceived the study. LR designed and performed the research, and wrote the original manuscript. L.R., T.I. and J.M. contributed to the scientific discussion and the revision of the paper.

## Funding

## Competing interests

The authors declare no competing interests.
