## [Peer Review File · Nature Communications]

Moderate greenhouse climate and rapid carbonate formation after Marinoan snowball EarthREVIEWER COMMENTS

Reviewer #1 (Remarks to the Author):

Review of "Moderate Greenhouse climate and rapid carbonate formation after Marinoan snowball Earth" by Ramme et al., submitted to Nature Communications Aug 2023

This is an interesting study that models the short-term (5 thousand year) evolution of the ocean-atmosphere carbon cycle following several different possible states at the moment of deglaciation following a snowball earth event. The paper convincingly makes the case that depending on the initial state chosen, atmospheric CO₂ could go up, down, or sideways depending on which feedback mechanism dominates.

My main critique is that the paper seems to accept as unquestionable dogma a few aspects of the sedimentary record of the marinoan caps that are actually disputed. Most importantly the timescale of cap carbonate deposition (see points below) and the interpretation of pCO₂ from geochemical records of the caps. My recommendation is to simply tone down those claims, or to put them within conditional statements e.g. "IF we take the position of Hoffman and others that the cap carbonates were deposited within thousands of years, THEN ..."

Line 231: It is unclear to me how freshening of seawater (by itself) would lead to increased carbonate mineral saturation state (omega) and accelerated carbonate precipitation and burial. If the only parameter being changed is salinity then yes, decreasing salinity causes an increase in omega. But during a deglaciation, the influx of freshwater dilutes not just salinity but also alkalinity, and the pre-existing [DIC]. When you decrease salinity, DIC, and alkalinity (by the same dilution factor), you get a pronounced decrease in omega. I did some very simple calculations using a carbonate system calculator to demonstrate this:

Salinity=35, Temp=25, Alk=2200, DIC=2000. Calculated Omega(calcite) = 3.54

(below: only salinity is reduced by half)

Salinity=17.5, Temp=25, Alk=2200, DIC=2000. Calculated Omega(calcite) = 4.55

(below: salinity, alkalinity, DIC all reduced by half)

Salinity=17.5, Temp=25, Alk=1100, DIC=1000. Calculated Omega(calcite) = 1.87

Line 239: The rapid deposition of Marinoan cap carbonates (within thousands of years) not as certain as the authors state. It is disputed from the basis of long-term carbon-cycle modeling (e.g. Penman & Rooney, 2018) and also the existence of paleomagnetic reversals within cap carbonates (Li, 2000, Trindale et al., 2003, Raub and Evans, 2005, summarized by Font et al., 2010 P3.). Together, these argue that the duration of cap carbonate deposition was more like many hundreds of thousands of years to more than a million years.

Figure 3: The duration of carbonate deposition in Figure 3c seems inconsistent with the fact that marinoan cap carbonates contain magnetic reversals, which typically occur on hundred-thousand year timescales.

Lines 255-262: seems like a repeat of what was said in lines 175-218

Figure 4: The authors seem to exclude scenarios of low initial ocean alkalinity because they are "too acidic to allow for a rapid formation of Marinoan cap dolostone" but as explained above, the rapid formation of the caps is not certain. It seems to me that scenarios of low initial ocean alkalinity are consistent with cap dolostones deposited over hundreds of thousands of years, which is consistent with the paleomagnetic reversals preserved therein.

Line 305: I would not treat the CO₂ concentration limits supposed by the Kasemann (2005) boron isotope study as very robust. There is so much uncertainty about boron isotopes in ~600 million year old inorganically precipitated carbonate (see Stewart et al 2015 Geology)

Reviewer #2 (Remarks to the Author):

This manuscript is one of the few that attempt to better understand the role of ocean's carbon cycle and in the aftermath of the latter of the two Neoproterozoic pan-glacials (the Marinoan snowball) using a full-scale Earth System Model (ESM). The authors claim that multiple rapid mechanisms related to the ocean's carbon cycle can substantially affect the atmospheric CO₂ in the aftermath of the Marinoan snowball and quantify it. This is indeed an important research topic that needs to be explored in depth with state-of-the-art ESMs. The paper is quite well written and does take into account some of the differences between the modern and the Neoproterozoic climates into account (e.g., the different continental arrangements during the two era). However, the authors make several incorrect assumptions regarding the initial conditions of the simulations and the conclusions drawn based on these simulations may be misleading if not completely incorrect. Thus, I cannot recommend the manuscript for publication in its current state. Please find my detailed comments in the file attached.

Reviewer's comment

Author's response

- *Change in manuscript (given line numbers refer to the revised manuscript)*

Anonymous Reviewer #1

Review of “Moderate Greenhouse climate and rapid carbonate formation after Marinoan snowball Earth” by Ramme et al., submitted to Nature Communications Aug 2023

This is an interesting study that models the short-term (5 thousand year) evolution of the ocean-atmosphere carbon cycle following several different possible states at the moment of deglaciation following a snowball earth event. The paper convincingly makes the case that depending on the initial state chosen, atmospheric CO₂ could go up, down, or sideways depending on which feedback mechanism dominates.

My main critique is that the paper seems to accept as unquestionable dogma a few aspects of the sedimentary record of the marinoan caps that are actually disputed. Most importantly the timescale of cap carbonate deposition (see points below) and the interpretation of pCO₂ from geochemical records of the caps. My recommendation is to simply tone down those claims, or to put them within conditional statements e.g. “IF we take the position of Hoffman and others that the cap carbonates were deposited within thousands of years, THEN ...”

We thank the reviewer for their comments. We agree that the original version of the manuscript did not really discuss the possibility that cap dolostones have been deposited on a much longer time scale. We therefore added a discussion of the existing discrepancy in the data. Since the mechanism we present in this work is indeed only relevant for scenarios of rapid cap dolostone deposition, we adopted the proposed conditional statements at the relevant locations in the text.

- *Added a part that discusses the contradicting evidence (lines 379-388)*

- *changed wording at several locations in the text (lines 406-407, 456-457, 461, 479)*

Line 231: It is unclear to me how freshening of seawater (by itself) would lead to increased carbonate mineral saturation state (omega) and accelerated carbonate precipitation and burial. If the only parameter being changed is salinity then yes, decreasing salinity causes an increase in omega. But during a deglaciation, the influx of freshwater dilutes not just salinity but also alkalinity, and the pre-existing [DIC]. When you decrease salinity, DIC, and alkalinity (by the same dilution factor), you get a pronounced decrease in omega. I did some very simple calculations using a carbonate system calculator to demonstrate this:

Salinity=35, Temp=25, Alk=2200, DIC=2000. Calculated Omega(calcite) = 3.54

(below: only salinity is reduced by half)

Salinity=17.5, Temp=25, Alk=2200, DIC=2000. Calculated Omega(calcite) = 4.55

(below: salinity, alkalinity, DIC all reduced by half)

Salinity=17.5, Temp=25, Alk=1100, DIC=1000. Calculated Omega(calcite) = 1.87

The reviewer is correct that the overall dilution will of course reduce the saturation state. What was meant here was indeed the isolated effect of lower salinity on the saturation state, but we agree that the way it was written in the manuscript was confusing and not representing properly what is happening during the deglaciation; we therefore modified the sentence. Furthermore, we also moved the discussion of the dilution effect from the methods to the main text (see "General modification"), which should also make this more clear.

- *modified the sentence around lines 328-331 to: "When the snowball Earth deglaciates, the saturation state of carbonates initially decreases due to the dilution of tracers during the inflow of meltwater, but subsequently increases again due to the strong warming of the ocean. Additionally, biological activity reduces $p\text{CO}_2$ in the surface ocean, further increasing the carbonate saturation state."*

Line 239: The rapid deposition of Marinoan cap carbonates (within thousands of years) not as certain as the authors state. It is disputed from the basis of long-term carbon-cycle modeling (e.g. Penman & Rooney, 2018) and also the existence of paleomagnetic reversals within cap carbonates (Li, 2000, Trindale et al., 2003, Raub and Evans, 2005, summarized by Font et al., 2010 P3.). Together, these argue that the duration of cap carbonate deposition was more like many hundreds of thousands of years to more than a million years.

Penman and Rooney (2018) also acknowledge the possibility that higher ocean alkalinity could induce more immediate cap carbonate deposition, which is what we simulate here. As stated above, the manuscript now includes a discussion of the issue about magnetic reversals with the cap carbonates, and we made our statements more conditional to the case of rapid carbonate formation.

Figure 3: The duration of carbonate deposition in Figure 3c seems inconsistent with the fact that marinoan cap carbonates contain magnetic reversals, which typically occur on hundred-thousand year timescales.

The issue of magnetic reversals was already discussed above; it is possible that these reversals occurred much more frequently at the time of the Marinoan snowball Earth, because the Earth's inner core had not yet solidified (see discussion in Hoffmann et al., 2017, last paragraph of section "Time scale for cap-carbonate transgressions" on page 31), but of course this is uncertain and debated, and we made our statements conditional to this.

Nevertheless, the rates we inferred from our calculations are indeed quite extreme and rapid, which is mostly a consequence of the settings of our simulation: we only simulate a comparably weak and short-lived dilution effect, biological activity resumes very strongly and quickly after the snowball Earth and no continental weathering flux is prescribed, which could prolong the carbonate deposition. The inferred rates are therefore indeed too rapid, but not by orders of magnitude. Our conclusion that the process of carbonate formation we describe in our manuscript could have contributed to a rapid ($< 10^4$ years) formation of Marinoan cap dolostones, is therefore still valid.

- *we largely reworked the discussion of the simulated carbonate deposition to better reflect the above-mentioned uncertainties and indicate that our results should really only be seen as first-order estimates (paragraph in lines 391-407)*

Lines 255-262: seems like a repeat of what was said in lines 175-218

Yes, this was meant as a introduction to a more general discussion, where we wanted to summarize what was important from the carbon cycle process analysis. Since in the original manuscript also this analysis was relatively short, this indeed seemed like a repetition. In the revised manuscript, we moved much more information on the carbon cycle process analysis from the methods into the main part (see "General modifications"), so that now this summary at the start of the discussion is much more sensible in our eyes.

- *modified this part to better summarize the newly added information in the sections before (lines 410-424)*

Figure 4: The authors seem to exclude scenarios of low initial ocean alkalinity because they are “too acidic to allow for a rapid formation of Marinoan cap dolostone” but as explained above, the rapid formation of the caps is not certain. It seems to me that scenarios of low initial ocean alkalinity are consistent with cap dolostones deposited over hundreds of thousands of years, which is consistent with the paleomagnetic reversals preserved therein.

We did not mean to completely exclude these cases, but the way it was presented in the original figure may have led to that conclusion. We therefore improved the quality of Fig. 4. It now specifically mentions that at low initial alkalinity, the carbonates could only be formed on those longer time scales.

- *updated Fig. 5 (previously Fig. 4)*

Line 305: I would not treat the CO₂ concentration limits supposed by the Kasemann (2005) boron isotope study as very robust. There is so much uncertainty about boron isotopes in ~600 million year old inorganically precipitated carbonate (see Stewart et al 2015 Geology)

We thank the reviewer for this assessment and modified the wording to better reflect this uncertainty.

- *modified sentence in line 461*

General modifications (unspecific to the reviewer)

- We moved the description of how we quantified the individual carbon cycle processes from the Methods into the main part of the manuscript. In the main part there are now subsections which shortly discuss the functioning of the individual carbon cycle processes within the pre-existing section "*How the ocean modulates atmospheric CO₂*". This was done because these descriptions were originally only moved to the method section, in order to keep the main part short and concise, fitting with stricter manuscript length requirements. In Nature Communications, however, the main part is allowed to be longer, and, also based on the reviewers comments, we feel that adding these descriptions back into the main part improves the quality of the manuscript. The descriptions provide a better explanation of how the carbon cycle functions directly in the main part of the manuscript and not just in the Methods. We note that no new information has entered the manuscript during this restructuring. The text was just moved from the Methods to the main part and adapted slightly to comply with the flow of the text.
- We added a figure to the Extended data that shows the impact of the dilution effect on the carbon cycle, as inferred from the idealized box model calculations. This information was previously only summarized in the text and we felt it would be appropriate to show the actual results somewhere. We also changed the order of figures in the Extended data to fit with how they are referred to in the text.

Reviewer's comment

Author's response

- Change in manuscript (given line numbers refer to the revised manuscript)

Anonymous Reviewer #2

Review of “Moderate greenhouse climate and rapid carbonate formation after Marinoan snowball Earth”

Summary

This manuscript is one of the few that attempt to better understand the role of ocean's carbon cycle and in the aftermath of the latter of the two Neoproterozoic pan-glacials (the Marinoan snowball) using a full-scale Earth System Model (ESM). The authors claim that multiple rapid mechanisms related to the ocean's carbon cycle can substantially affect the atmospheric CO₂ (pCO₂) in the aftermath of the Marinoan snowball and quantify it. This is indeed an important research topic that needs to be explored in depth with state-of-the-art ESMs. The paper is quite well written and does take into account some of the differences between the modern and the Neoproterozoic climates into account (e.g., the different continental arrangements during the two era). However, the authors make several incorrect assumptions regarding the initial conditions of the simulations and the conclusions drawn based on these simulations may be misleading if not completely incorrect. Thus, I cannot recommend the manuscript for publication in its current state. Please find my detailed comments below.

We thank the reviewer for the positive feedback on the importance and novelty of our work and address the reviewer's concern about incorrect assumptions and our conclusions point by point in the comments below. Overall, we are grateful for the constructive criticism, as we think that the additional simulation and the modifications we made have improved the robustness of our conclusions.

Major comments

A brief examination of Ilyina et al. (2013) and the adjustments made in this study indicates that the following processes are expected affect [DIC] and [T A] in the Neoproterozoic ocean: a) evaporation/precipitation/freshwater fluxes, b) primary production, c) organic matter remineralization and d) air-sea flux of CO₂. Further, primary production depends on both the nutrient availability and temperature (T; for T < 35 °C). Additionally, the treatment of carbonate chemistry in the model is such that abiotic carbonate formation is said to occur when $\Omega_{\text{CaCO}_3} > 15$, where $\Omega_{\text{CaCO}_3} \propto [\text{CO}_3^{2-}]$.

1. The authors have been honest about using a highly debated initial [PO₄³⁻] (~ 5 times higher than the modern value; Planavsky et al., 2010). Previous studies show that a low O₂ atmosphere during the Proterozoic could only be sustained in a [PO₄³⁻] deficient ocean (Laakso and Schrag, 2017; Reinhard et al., 2017). Given how strongly the ocean biology affects [DIC] and [T A] in the ocean, any conclusions that are drawn based on these simulations can be misleading (if not outright incorrect) since it is not trivial to determine whether the conclusions of this study will still be true for lower values of [PO₄³⁻].

The reviewer correctly points out that our choice of very high [PO₄³⁻] is questionable. However, the references the reviewer cites also acknowledge that there was a transition to higher phosphate and

oxygen concentrations at some point during the late Neoproterozoic. While Laakso and Schrag (2017) indeed discuss that a low oxygen atmosphere during the Proterozoic could only be sustained at low $[\text{PO}_4^{3-}]$, they also show that phosphate and oxygen accumulated during the Neoproterozoic snowball Earth events, so that specifically the aftermath of the Marinoan snowball Earth, which we study here, may indeed have already seen higher oxygen and phosphate concentrations. Reinhard et al. (2017) state that phosphate was low until about 800 to 700 million years ago (Ma), but they also state that a fundamental shift in the phosphorous cycle may have occurred between 800 and 635 Ma. The aftermath of the Marinoan snowball Earth which we simulate is situated at 635 Ma, i.e. after this potential shift. Therefore, it not at all certain that $[\text{PO}_4^{3-}]$ and $[\text{O}_2]$ must have been very low, but we agree that we use a very high concentration of phosphate and that $[\text{PO}_4^{3-}]$ could have been much lower, which should be discussed in our manuscript.

In order to address this issue, we designed an additional simulation with a very low initial $[\text{PO}_4^{3-}]$ of $10^{-4} \text{ mol m}^{-3}$, accompanied by a low initial oceanic $[\text{O}_2]$ of $10^{-5} \text{ mol m}^{-3}$ and a reduced atmospheric O_2 concentration of 0.1 PAL (from 0.5 PAL in the other runs). Apart from that, the simulation is identical to one of the pre-existing simulations so that the effect of low $[\text{PO}_4^{3-}]$ and oxygen can be studied in isolation. We discuss the new simulation in the section on the general effect of the biological activity on the carbon cycle. The following two figures show some time series of the new simulation (Exp. 3.3) and the reference simulation (Exp. 3.2). Figure 1 was also added to the manuscript, as we think that it not just shows the impact of $[\text{PO}_4^{3-}]$ availability on the carbon cycle, but also provides an illustrative value by indicating the effects of the different carbon cycle processes.

Fig. 1: Air-sea CO₂ exchange in experiments with high (Exp. 3.2) and low (Exp. 3.3) oceanic phosphate concentration.

Figure 2 shows the impact of low $[\text{PO}_4^{3-}]$ on biological activity and thereby carbonate formation. While in the old Exp. 3.2 net primary production (npp) was divided between cyanobacteria (60%) and phytoplankton (40%), the much reduced npp in Exp. 3.3 is completely based on phytoplankton. As npp is much lower here, the biological carbon pump (BCP) is also much weaker, leading to less transport of carbon into the deep ocean and hence a higher surface pCO₂. This initially leads to the much stronger outgassing of CO₂ during the first ~1000 years from the start of the deglaciation. However, this also reduces the saturation state of carbonate and weakens the rapid carbonate formation in our simulations substantially, leading to less long-term outgassing of CO₂. These two processes, the BCP and biology-

induced carbonate formation, roughly balance each other out and the atmospheric CO₂ concentration is almost identical at the end of the two simulations. This means that in scenarios where the ocean carbonate saturation state is generally high enough to allow for inorganic formation of carbonates, there is a direct positive relationship between biological activity and that part of carbonate formation that is induced by biological activity (there is still also the warming-induced component of carbonate formation).

Fig. 2: Net primary production and inorganic carbonate formation in experiments with high (Exp. 3.2) and low (Exp. 3.3) oceanic phosphate concentration.

In summary, the lower [PO₄³⁻] leads to a much reduced primary production in the ocean and hence a much weaker BCP. This has two main impacts:

- The weaker BCP means there is much less oceanic CO₂ uptake during the period of reviving biological activity. This means that the initial reduction in atmospheric CO₂ that we see in the other simulations is much weaker or even reversed, as other processes then dominate.
- The weaker BCP also means that Ω_{CaCO_3} is lower in the surface ocean (because pCO₂ is higher, as less DIC is transported into the deep ocean). This reduces the inorganic formation of carbonates, leading to less long-term outgassing of CO₂. The reduced outgassing due to less carbonate formation actually balances the reduced uptake due to a weaker BCP, so that the net effect of lower [PO₄³⁻] on atmospheric CO₂ is very small.

Overall, the new simulation has enhanced our understanding of the impact of biology on the carbon cycle after the Marinoan snowball Earth and we are thankful that the reviewer made this comment. The new simulation has shown that biological activity is probably the most uncertain carbon cycle process during that time, and that its impact on atmospheric CO₂ can vary from no impact (if there is low primary production and the BCP-related CO₂ uptake is balanced by carbonate formation inducing CO₂ outgassing) to a reduction of atmospheric CO₂ by up to 2000 ppm (if there is high primary production and no balancing at low Ω_{CaCO_3}). However, all other carbon cycle processes that we discuss in the manuscript are mostly unaffected, so that the uncertainty in [PO₄³⁻] actually only adds to the uncertainty in one of the five carbon cycle processes we discuss in our manuscript. Most importantly, the effect of biology on the carbon cycle is not dominating the evolution of the supergreenhouse climate. In fact, it is possibly even the weakest of those carbon cycle processes. The large shifts in the physical and

chemical conditions of the ocean during the snowball Earth aftermath, i.e. the excessive warming, the inflow of huge amounts of meltwater, the possible large disequilibrium between atmosphere and ocean and the amount of alkalinity in the ocean at the start of the deglaciation, are dominating over the effect of biology on the time scale we discuss in our manuscript. It is true that biology probably had a major impact on the long-term evolution of the Earth system during the late Neoproterozoic, but during the first five thousand years after the snowball Earth, the physical and chemical transformations are far more important than biology. For this reason our conclusions are robust with respect to the impact of a lower $[\text{PO}_4^{3-}]$, but of course we include everything that we have learned from the new simulation into the manuscript.

- *the new simulation was integrated into the existing figures and tables (Figs. 1, S1, S2, Tab. S1)*
- *we added a new figure that shows the impact of the adapted $[\text{PO}_4^{3-}]$ and $[\text{O}_2]$ on the air-sea CO_2 exchange compared to a reference simulation (Fig. 3)*
- *In the description of the carbon cycle processes we now have a section about the impact of biology, in which also the availability of nutrients is discussed (lines 235-278)*
- *the section describing the process of carbonate formation was modified to better reflect what we have learned through the new simulation (lines 320-359)*
- *wherever in the original manuscript we spoke of "warming-induced carbonate formation", we adapted the wording to better reflect that also biology has an impact on carbonate formation*
- *updated Figs. 2 and 5 to better represent the new information.*

2. Using an initial seawater $[\text{PO}_4^{3-}] = 10^{-2} \text{ mol m}^{-3}$ would yield an exceptionally productive ecosystem in the ocean, since the half saturation constant for PO_4^{3-} -uptake is $10^{-5} \text{ mol m}^{-3}$. Assuming a more realistic $[\text{PO}_4^{3-}]$, say between 1-10% of modern-levels (2.6×10^{-5} to $2.6 \times 10^{-4} \text{ mol m}^{-3}$), yields a primary production that is anywhere between 4–30% less than what the authors obtain, which directly affects the [DIC] at the surface and hence the rate at which pCO_2 changes via air-sea exchange. Given that the model has an interactive carbon cycle, I suspect that using a realistic $[\text{PO}_4^{3-}]$ will change the evolution of pCO_2 . This in turn will affect the surface air temperature and hence can alter the temperature sensitive CO_3^{2-} precipitation rates. Do the authors expect CO_3^{2-} precipitation rates be the same at lower values of $[\text{PO}_4^{3-}]$?

Indeed, in our simulations we have very high primary production, which is largely reduced at lower $[\text{PO}_4^{3-}]$ and this difference impacts the evolution of atmospheric CO_2 and the carbonate precipitation rates, as we show above. However, the effect on the overall carbon cycle is dominated by the warming-induced part of carbonate precipitation, as the CO_2 outgassing of the biology-induced carbonate formation is balanced by the CO_2 uptake of the BCP. It is the effect on the carbon cycle and atmospheric CO_2 , which we focus on in the manuscript. We do not try to give precise estimates of carbonate precipitation rates, due to the many uncertainties and the model simplifications. With respect to carbonate formation, our aim in this paper is to show two things: 1. Rapid carbonate formation in the snowball Earth aftermath can lead to substantial outgassing of CO_2 . 2. Rapid carbonate formation

induced by ocean warming and biological activity could explain in part the Marinoan cap dolostones. We think that the revised manuscript does this, and we put even more emphasis on the fact that the inferred carbonate precipitation rates could also be different in reality and that our results only indicate a potential impact of this carbon cycle process.

- *Reworked the section about carbonate formation and the deposition of Marinoan cap dolostones (lines 378-407).*

3. Prescribing such high $[\text{PO}_4^{3-}]$ would also result in an excessive amount of dissolved organic matter (DOM) and detritus that will be remineralized aerobically. Aerobic remineralization of DOM and detritus in an oxygen rich ocean (due to PO_4^{3-} abundance) would in itself be a rapid process that increases [DIC] in the ocean. In fact, the parameterization in HAMOCC is such that aerobic remineralization of detritus is 1.25 times faster than anaerobic remineralization (Ilyina et al., 2013). This rapid restoration of [DIC] due to remineralization of excessive of DOM and detritus would swiftly increase the CO_3^{2-} and can yield a high Ω_{CaCO_3} , which in turn can drive a rapid abiotic carbonate formation that we see in the simulations described here. How do the authors disentangle the effect of ocean biology from abiotic precipitation of carbonates?

Remineralisation of DOM (or detritus) can only release as much [DIC] as there was previously taken up during photosynthesis, and the whole process of biological carbon uptake at the surface and release in the deep ocean is what we discuss in the section about the BCP. An increase of [DIC] due to remineralisation does actually not increase Ω_{CaCO_3} , because most importantly it reduces pH, despite a possible increase in $[\text{CO}_3^{2-}]$. Even if it would, remineralisation often occurs not at the surface, but in deeper layers, where the pressure effect reduces the carbonate saturation state and no carbonate formation occurs. Therefore, the remineralisation of DOM has no impact on the abiotic carbonate precipitation, and we already discuss its impact on the carbon cycle in the section about the BCP. The only other way that DOM remineralisation could impact the carbon cycle in the snowball aftermath would be, if there existed a large DOM reservoir in the ocean at the start of the deglaciation, which is remineralised during deglaciation, leading to outgassing. But this is not the case. In fact, the DOM concentration is several orders of magnitude smaller at the end of the deglaciation than during our control climate, because there is effectively no primary production during the snowball state. DOM only starts to build up in the snowball Earth aftermath as a consequence of biological activity, which we discuss in the manuscript.

4. The authors also prescribe a very high initial O_2 , which would favour aerobic remineralization over anaerobic remineralization. This can lead to erroneous estimates in the rate of CO_3^{2-} precipitation (see last paragraph of major comment #3).

We also account for lower oxygen concentrations in the new experiment, and as we discussed above, remineralisation has no major impact on the rate of carbonate formation. The differences we see for carbonate formation in the new simulation are mostly a consequence of the reduced BCP strength due to lower $[\text{PO}_4^{3-}]$ and not because of the reduced oxygen concentration.

In light of these critical issues. I would recommend that the authors consider redoing the simulations with the correct initial conditions. Ideally, the biogeochemical component of the model should also

account for the evolution of atmospheric O₂ to determine how the oceanic O₂ evolves. This will ensure that the CO₃²⁻ precipitation rates are estimated correctly.

As we discuss above, we do not agree that our high values of [PO₄³⁻] and [O₂] are for certain completely incorrect. Instead they are just at the very high end of an uncertainty spectrum. Since our simulation strategy was anyhow to cover all possibilities by simulating the extremes at both ends of an uncertainty range, we are thankful that the reviewer showed us that we do not cover the uncertainty range of phosphate and oxygen concentrations. We now have run a simulation with much lower [PO₄³⁻] and [O₂], to address the possibility that these concentrations were not as high as in our original simulations. The new simulation is incorporated and discussed in the new version of the manuscript, and we now have a full section about the impact of biology on the carbon cycle. The main outcome is, however, that also a very different biological system does not change the major evolution of the carbon cycle, which is mostly driven by the other (abiotic) processes like the rapid warming, the dilution effect or the equilibration of atmosphere and ocean. Therefore, our conclusions are still valid and we do not see the need to repeat all simulations. Furthermore, accounting for the evolution of O₂ is not possible with our model, and simulating the coupled evolution of oxygen and the snowball Earth glacial cycle would constitute a whole new study on its own. The focus of our work is the carbon cycle, and the availability of oxygen does not affect the carbon cycle by much on the time scale that we discuss in our work.

Minor comments

1. The degree of disequilibrium between the atmosphere and the ocean will be determined by the meridional extent of marine ice cover during the Marinoan pan glacials. The precise marine extent of ice cover during the Neoproterozoic pan-glacials is yet to be determined. This needs to be emphasized further.

This is correct, and some of our simulations were specifically designed to address this uncertainty in the magnitude of the process, which we call "reservoir equilibration effect". We modified a sentence to indicate that it is this specific distinction between "hard" and "soft" snowball Earth that we are thereby accounting for. Additionally, as we moved a more detailed description from the method section into the main text, this should have become even clearer.

- *added a sentence in lines 72, 86-88*

- *moved the description of this process from the methods to the main text (see "General modifications")*

2. Although this isn't something related to the science, I would recommend that the authors use a margin of 1 inch and line spacing of 1.5 pts when submitting a manuscript. The authors' current choice of styling does not appropriately showcase the otherwise very aesthetic figures.

We adapted the settings in the new version of the manuscript and thank the reviewer for this advise.

General modifications (unspecific to the reviewer)

- We moved the description of how we quantified the individual carbon cycle processes from the Methods into the main part of the manuscript. In the main part there are now subsections which shortly discuss the functioning of the individual carbon cycle processes within the pre-existing section "*How the ocean modulates atmospheric CO₂*". This was done because these descriptions were originally only moved to the method section, in order to keep the main part short and concise, fitting with stricter manuscript length requirements. In Nature Communications, however, the main part is allowed to be longer, and, also based on the reviewers comments, we feel that adding these descriptions back into the main part improves the quality of the manuscript. The descriptions provide a better explanation of how the carbon cycle functions directly in the main part of the manuscript and not just in the Methods. We note that no new information has entered the manuscript during this restructuring. The text was just moved from the Methods to the main part and adapted slightly to comply with the flow of the text.
- We added a figure to the Extended data that shows the impact of the dilution effect on the carbon cycle, as inferred from the idealized box model calculations. This information was previously only summarized in the text and we felt it would be appropriate to show the actual results somewhere. We also changed the order of figures in the Extended data to fit with how they are referred to in the text.

REVIEWERS' COMMENTS

Reviewer #1 (Remarks to the Author):

I felt that my concerns were adequately addressed

Reviewer #2 (Remarks to the Author):

Review of "Moderate greenhouse climate and rapid carbonate formation after Marinoan snowball Earth" by Ramme et al.

Having read the revised version of the manuscript and the authors' response to the reviewers' comments, I find that the authors have successfully addressed the concerns that were raised in the original review. In particular, the inclusion of a simulation with a low seawater phosphate concentration and a separate section that discusses the biological carbon pump has added more depth to the study. I do not have any other comments/suggestions, and I believe that the manuscript is now fit for publication.